# Intrinsic protein disorder is insufficient to drive subnuclear clustering in embryonic transcription factors

**Colleen E Hannon\*, Michael B Eisen**

Howard Hughes Medical Institute, University of California, Berkeley, United States

**Abstract** Modern microscopy has revealed that core nuclear functions, including transcription, replication, and heterochromatin formation, occur in spatially restricted clusters. Previous work from our lab has shown that subnuclear high-concentration clusters of transcription factors may play a role in regulating RNA synthesis in the early *Drosophila* embryo. A nearly ubiquitous feature of eukaryotic transcription factors is that they contain intrinsically disordered regions (IDRs) that often arise from low complexity amino acid sequences within the protein. It has been proposed that IDRs within transcription factors drive co-localization of transcriptional machinery and target genes into high-concentration clusters within nuclei. Here, we test that hypothesis directly, by conducting a broad survey of the subnuclear localization of IDRs derived from transcription factors. Using a novel algorithm to identify IDRs in the *Drosophila* proteome, we generated a library of IDRs from transcription factors expressed in the early *Drosophila* embryo. We used this library to perform a high-throughput imaging screen in *Drosophila* Schneider-2 (S2) cells. We found that while subnuclear clustering does not occur when the majority of IDRs are expressed alone, it is frequently seen in full-length transcription factors. These results are consistent in live *Drosophila* embryos, suggesting that IDRs are insufficient to drive the subnuclear clustering behavior of transcription factors. Furthermore, the clustering of transcription factors in living embryos was unaffected by the deletion of IDR sequences. Our results demonstrate that IDRs are unlikely to be the primary molecular drivers of the clustering observed during transcription, suggesting a more complex and nuanced role for these disordered protein sequences.

**\*For correspondence:**
cehannon@berkeley.edu

**Competing interest:** The authors declare that no competing interests exist.

## eLife assessment

The manuscript addresses a **fundamental** question: are IDRs responsible for subnuclear clustering of transcription factors? A screen of 75 IDRs yielded **convincing** evidence that IDRs are rarely sufficient for subnuclear clustering, while the experimental design and data analysis provided limited evidence for the authors' claims regarding transcription factor clustering.

## Introduction

Gene expression is a tightly regulated process that requires the coordinated assembly of transcriptional machinery at specific sites in the genome. This process depends on interactions between sequence-specific DNA-binding transcription factors (TFs) at enhancer sequences, general TFs, and co-activators including the Mediator complex, and RNA polymerase II (PolII) at promoters (*Reiter et al., 2017*). Classical models of transcription describe a mechanism in which TFs bound at distant enhancers are able to communicate with PolII at the promoter via chromatin looping (*Su et al., 1991*), and there is ample biochemical and genetic evidence for this process (*Kornberg, 2005*; *Krivega and Dean, 2012*; *Robinson et al., 2016*). Transcription is also highly dynamic, and must be regulated

tightly in both space and time in response to environmental stimuli and developmental changes. As a result, the transcriptional machinery must be capable of being rapidly assembled and disassembled within the nucleus. These real-time molecular dynamics cannot be captured by in vitro biochemical studies or genomic approaches that provide population-averaged data.

The interactions between regulatory DNA and the transcriptional machinery are determined in part by the molecular structure of TF proteins. Eukaryotic TFs share common structural components, having at minimum a sequence-specific DNA-binding domain and transcriptional activation domain (*Brent and Ptashne, 1985*; *Kadonaga et al., 1988*; *Keegan et al., 1986*). While the DNA-binding domains of TFs are typically folded into a defined structure, regions of the protein outside of the DNA-binding domain, including transcriptional activation domains, often remain flexible or unstructured (*Guo et al., 2012*). These intrinsically disordered regions (IDRs), which often have lower complexity in their amino acid sequence than structured domains, are highly overrepresented in eukaryotic TFs and essential for their function (*Liu et al., 2006*; *Minezaki et al., 2006*; *Staby et al., 2017*). However, the precise molecular function of IDRs and their broader role in regulating transcriptional activity remains unclear. Additionally, because they do not fold into a three-dimensional structure and are difficult to assess by traditional biochemical methods, IDRs have been understudied relative to structured protein domains.

Modern advances in imaging technologies have allowed observations of transcription at single loci in living cells and illuminated a highly dynamic process (*Coleman et al., 2015*; *Darzacq et al., 2007*; *Fukaya et al., 2016*; *Garcia et al., 2013*; *Zhang et al., 2016*). Single-molecule imaging of PolII, for example, shows that it forms transient clusters containing multiple active polymerases which are associated with active transcription at individual loci (*Cisse et al., 2013*). Biochemical studies suggest that localized regions of active transcription arise from interactions between the repeat-rich carboxy-terminal domain of the RBP1 subunit of PolII and the activation domains of TFs that contain IDRs with low sequence complexity. These IDRs can be polymerized into gel-like droplets, which can bind to PolII in vitro (*Kwon et al., 2013*).

Live imaging studies have also aided the characterization of disordered proteins in vivo, and, increasingly, evidence from these studies suggests that IDRs play a role in promoting interactions between TFs and the transcriptional machinery. The embryonic stem cell pluripotency factor Sox2 has been shown to form transient clusters at the enhancers of its target genes in living cells. It was proposed that cluster formation is mediated by weak protein-protein interactions between unstructured, low complexity sequences in the Sox2 activation domain, and that these clusters create high local concentrations of TFs and possibly associated co-factors to create hubs of gene activation (*Liu et al., 2014*). IDRs within TFs have also been demonstrated to mediate interactions with PolII in dynamic, high-concentration multi-protein hubs that activate transcription in human cells (*Chong et al., 2018*). PolII and Mediator have also been shown to form dynamic clusters in living cells that can co-localize within the nucleus (*Cho et al., 2018*). Biochemical evidence also demonstrates that Mediator can form phase-separated condensates with the human TFs OCT4 and GCN4 in vitro. The formation of these condensates requires the intrinsically disordered activation domains of the TFs, which are also necessary for transcriptional activation in vivo. These findings indicate that gene activation is driven by dynamic, low affinity interactions between TFs and co-factors at enhancers (*Boija et al., 2018*).

Taken together, these studies suggest that low sequence complexity IDRs may be mediating the formation of local high-concentration hubs of TFs and other transcriptional machinery at the enhancers of actively expressed genes. The formation of these hubs could mediate rapid transcriptional responses, and may be a method of organizing the nucleus into different functional domains. Work from our own lab has demonstrated that two maternally provided TFs, Bicoid and Zelda, form hubs of high local concentration within nuclei in developing *Drosophila* embryos. These hubs are highly dynamic and interact transiently with sites of transcription (*Mir et al., 2018*). Although both proteins are predicted to contain significant intrinsic disorder, with Zelda being particularly disordered (*Hamm et al., 2015*), it is unclear to what degree IDRs influence their dynamics in the nucleus.

Here, we interrogate the role of IDRs in driving TF cluster formation and nuclear organization. We hypothesized that IDRs are responsible for mediating the subnuclear localization and interactions of TFs and ultimately their function in transcriptional activation. Therefore, dissecting the role of IDRs in nuclear organization is essential for understanding the molecular logic of transcriptional regulation.

We sought to conduct a broad survey of tracts of intrinsic protein disorder within *Drosophila* TFs, assessing their contributions to the subnuclear dynamics that we observe in the early embryo. We identified IDRs from the *Drosophila* proteome and conducted a functional imaging screen of IDR localization in cultured cells, which demonstrated that IDRs are insufficient to drive subnuclear clustering on their own. Further structure-function analysis of IDR-containing TFs in live *Drosophila* embryos revealed that IDRs are dispensable for subnuclear clustering of TFs. The IDRs identified in this study are therefore neither necessary nor sufficient for the subnuclear clustering that we observe for full-length TFs, providing evidence against a model that they are the primary driver of multivalent interactions at sites of transcriptional activation.

## Results

### A hidden Markov model to predict IDRs from the proteome

In order to examine the in vivo behavior of IDRs, we first needed a high-throughput method to identify them in the proteome. Numerous protein disorder predictors exist, and there has been great progress in computationally predicting IDRs in recent years (*Liu et al., 2019*; *Necci et al., 2021*). Despite the ever-growing number of tools to predict disorder, at the outset of our study, we found that widely used predictors commonly output the probability that a given residue is located in an IDR, rather than the discrete coordinates that are reported for structured domains. Therefore, we sought to identify high-confidence IDRs in the *Drosophila* proteome by converting these scores into a set of coordinates.

We started with an existing disorder predictor, IUPred, which predicts disordered regions from an amino acid sequence by estimating the ability of polypeptides to form stabilizing contacts (*Dosztányi et al., 2005*; *Mészáros et al., 2018*). We applied the IUPred long algorithm to all amino acid sequences in the *Drosophila* proteome and queried the IUPred scores in regions annotated as known structured domains, as well as those in unannotated regions (*Figure 1A*). We found that, as expected, IUPred scores were lower (indicating less predicted disorder) in structured domains than in sequences outside of these domains (*Figure 1B*). We then implemented a simple hidden Markov model (HMM), using the proteome-wide IUPred scores as an input, to predict regions of structure and disorder. We fit the HMM using the annotated structured domains and unannotated regions of the proteome as negative and positive examples of disorder, respectively. We fit the HMM to a Viterbi prediction with binary outputs, and used the Viterbi prediction of the trained HMM to define a list of predicted structured (with a score of 0) regions and unstructured regions (with a score of 1) in the proteome. An example of the outputs from the HMM is shown in *Figure 1A*, for the daughterless protein. This protein is known to contain a basic helix-loop-helix DNA-binding domain (*Gebali et al., 2019*; *Letunic et al., 2015*; *Murre et al., 1989*), and our HMM successfully predicts a structured region corresponding to this domain and three continuous unstructured regions in the rest of the protein.

To assess the overall predictive power of our HMM, we compared the presence of a given amino acid in our Viterbi calls to those regions annotated as structured domains by Pfam/SMART. There was a strong enrichment for predicted structured sequence from our model in known Pfam/SMART domains and a strong depletion of predicted unstructured sequence in the same domains, relative to the rest of the proteome (*Figure 1C*). This result demonstrates that our HMM can successfully predict disorder from the proteome while largely excluding structured domains. Additionally, our Viterbi calls generally agree with existing annotations of IDRs, when checked against the MobiDB protein disorder database (*Di Domenico et al., 2012*; *Coronado et al., 2021*).

Using this list of structured and unstructured regions, we sought to generate a list of candidate IDRs to assess experimentally. We first took all of the unstructured domains and filtered for those found within TFs (*Shazman et al., 2014*) and known to be expressed in the early embryo (*Lott et al., 2011*). We further filtered the list for IDRs by size and genomic position in order to identify those that would be feasible to clone into a plasmid library. Using these filters, we generated a list of 78 IDRs contained within 72 unique TFs (*Supplementary file 1*).

### S2 cell screen for subnuclear clustering of IDRs

With our list of IDRs in hand, we designed a high-throughput functional imaging screen to assess their sufficiency in driving subnuclear localization. We generated a plasmid library containing each IDR tagged with mNeonGreen. We were unable to amplify 3 of the IDRs from the genome, so the

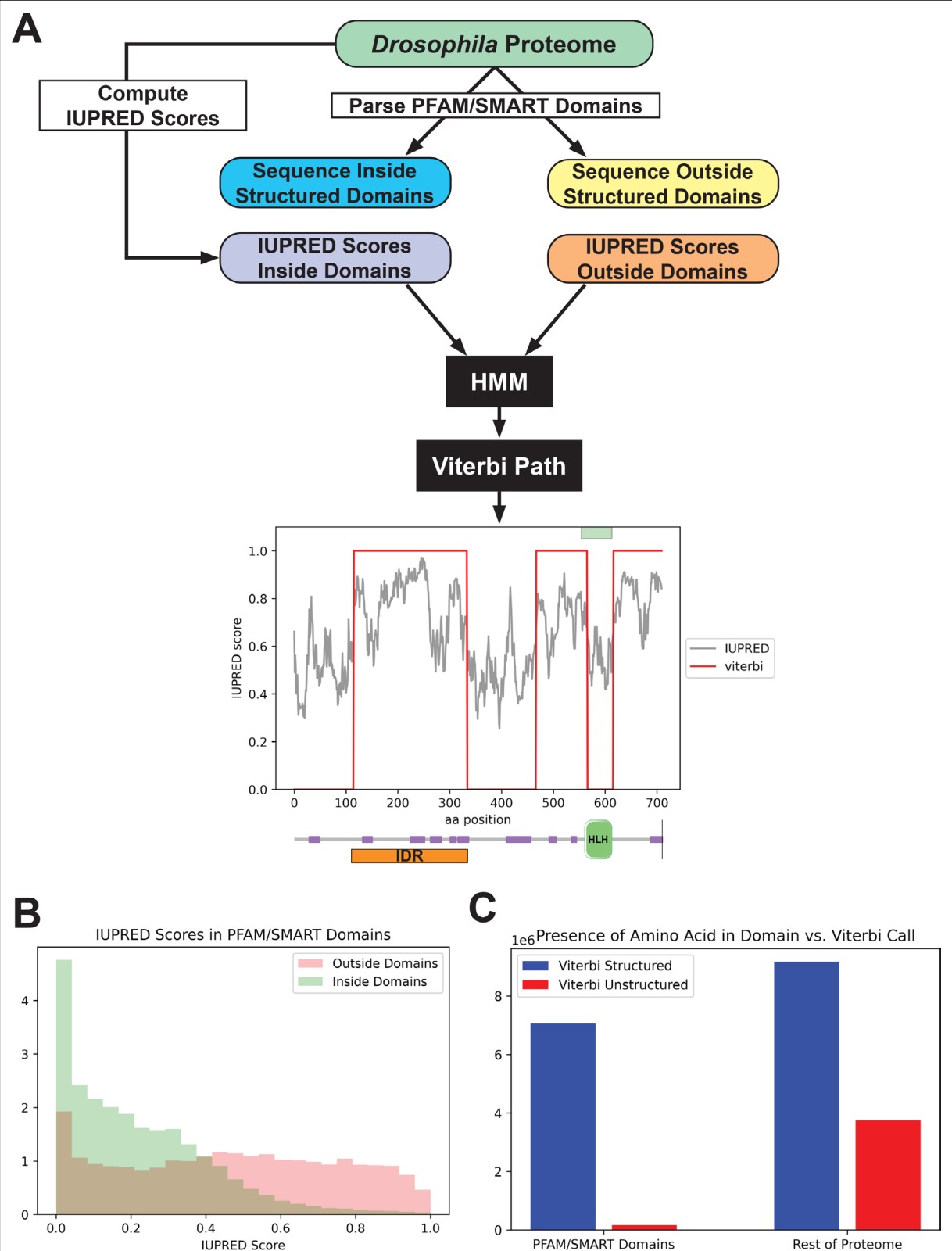

**Figure 1.** Predicting disorder from the proteome. (**A**) A summary of the workflow of the intrinsically disordered region (IDR) prediction algorithm. IUPred scores were computed for the entire proteome, and the output was parsed into scores for sequences inside annotated structured domains, or in unannotated regions of the proteome. These two sets of IUPred scores were used to train a hidden Markov model (HMM) to assign sequences in the proteome to 'structured' and 'unstructured' states. A Viterbi path was computed from the HMM to provide a binary output of the predictions. The plot

*Figure 1 continued on next page*

*Figure 1 continued*

shows the output of the algorithm for the daughterless transcription factor (TF). The IUPred 'long' scores are plotted in black, and the Viterbi path from our HMM is shown in red. The green box at the top of the figure denotes the annotated structured domain from SMART, extracted from the FlyBase GFF file, for this protein. Beneath the figure is a schematic of the linear protein structure (modified from SMART; *Letunic et al., 2015*; *Schultz et al., 1998*), with IDRs indicated in purple and a helix-loop-helix binding domain in green. The IDR isolated for this study is shown in orange. (**B**) Histogram showing the distribution of IUPred 'long' scores in regions of the proteome annotated as structured domains by Pfam and/or SMART (green) vs. regions outside of known domains (red). (**C**) The number of amino acids from the proteome that are classified as structured (blue) vs. unstructured (red) by our HMM Viterbi call in annotated Pfam/SMART domains and in regions of the proteome outside of known domains.

final library contained 75 IDRs. In order to rapidly screen the IDR library for clustering behavior, we conducted the first stage of our imaging using *Drosophila* Schneider-2 (S2) cultured embryonic cells. We used the plasmid library to generate stable cell lines expressing each IDR, along with a red (mRuby3) tagged Histone 2Av to mark the nucleus. High-throughput imaging of live cells revealed that nearly all of the IDRs were distributed homogeneously within the nucleus (*Figure 2A*), suggesting that IDRs alone are insufficient to drive subnuclear clustering behavior of the type that we observe in TFs in the embryo. While some IDRs show non-uniformity in the nucleus, very few resemble the fine-scale clusters that we observe for TF hubs in embryonic nuclei.

Two of the 75 IDRs were sufficient to drive clustering in S2 Cells: IDR16 from the protein MESR4 (*misexpression suppressor of ras 4*) and IDR72 from the protein Brk (*brinker*). Both IDRs show non-uniform nuclear distribution, with the MESR4 IDR having several large, bright clusters throughout the nucleus, and the Brk IDR showing more fine-scale clustering (*Figure 2B*). *MESR4* was first identified for playing a role in the RAS1 signaling pathway (*Huang and Rubin, 2000*) and has since been identified in several genetic screens, with roles in lipid metabolism and gene regulation (*Tsuda-Sakurai et al., 2015*), cellular responses to hypoxia (*Lee et al., 2008*), and germline differentiation (*Wissel et al., 2016*). The full-length protein contains nine C2H2-type zinc fingers and a plant homeodomain-type zinc finger (*Letunic et al., 2015*; *Schultz et al., 1998*). Interestingly, it was also identified as a component of the histone locus body (HLB), a nuclear compartment that serves as a site of histone synthesis (*White et al., 2011*). However, as the HLB appears as a single bright spot in the nucleus of S2 cells, HLB localization likely does not explain the clustering observed for the MESR4 IDR.

Brk is a transcriptional repressor that is negatively regulated by Dpp signaling in the embryo and the larval imaginal discs (*Jaźwińska et al., 1999*). The protein contains a Brinker DNA-binding domain, which has been shown to be unstructured and flexible in the absence of DNA, but to fold upon DNA-binding (*Cordier et al., 2006*). The DNA-binding domain is contained entirely within the IDR expressed in our study, which raises the possibility that the observed clustering of this IDR is the result of the tagged protein fragment binding to DNA in S2 cells.

Several of the IDRs also localize to different subnuclear or subcellular compartments (*Figure 2C*). For example, IDR13 is from the protein encoded by *CG42748*, which is predicted to be involved in the organization of cell-cell junctions (*Lye et al., 2014*), and localizes to both the nucleus and the plasma membrane. IDR31 appears to be strongly localized to the nucleolus. This IDR is from the largely uncharacterized protein encoded by *CG7839*, which contains a CCAAT binding factor domain (*Blum et al., 2021*). This domain is found in proteins known to play a role in 60S ribosomal subunit biogenesis in yeast (*Edskes et al., 1998*) and regulating *hsp70* expression in humans (*Lum et al., 1990*). The localization patterns observed in S2 cells indicate that some of the IDRs retain some degree of function or nuclear addressing that would be expected from the full-length protein. This result suggests that these particular IDRs play a functional role in protein localization.

## A subset of full-length TFs show strong subnuclear clustering in S2 cell nuclei

The nuclear uniformity of the majority of the IDRs in S2 cells suggested that in the majority of cases (73/75), IDRs alone are insufficient to drive the subnuclear clustering that we observe in fluorescently tagged TFs in the embryo. We next investigated whether full-length TFs showed clustering in S2 cells. We queried our list of IDRs and identified corresponding full-length proteins that were encoded by a single exon and could therefore easily be cloned into an expression plasmid for transfection into S2 cells. This gave us a list of nine TFs, shown in *Table 1*, for which we generated cell lines expressing the full-length proteins. IDR prediction plots for each of the TFs are shown in *Figure 3—figure supplement*

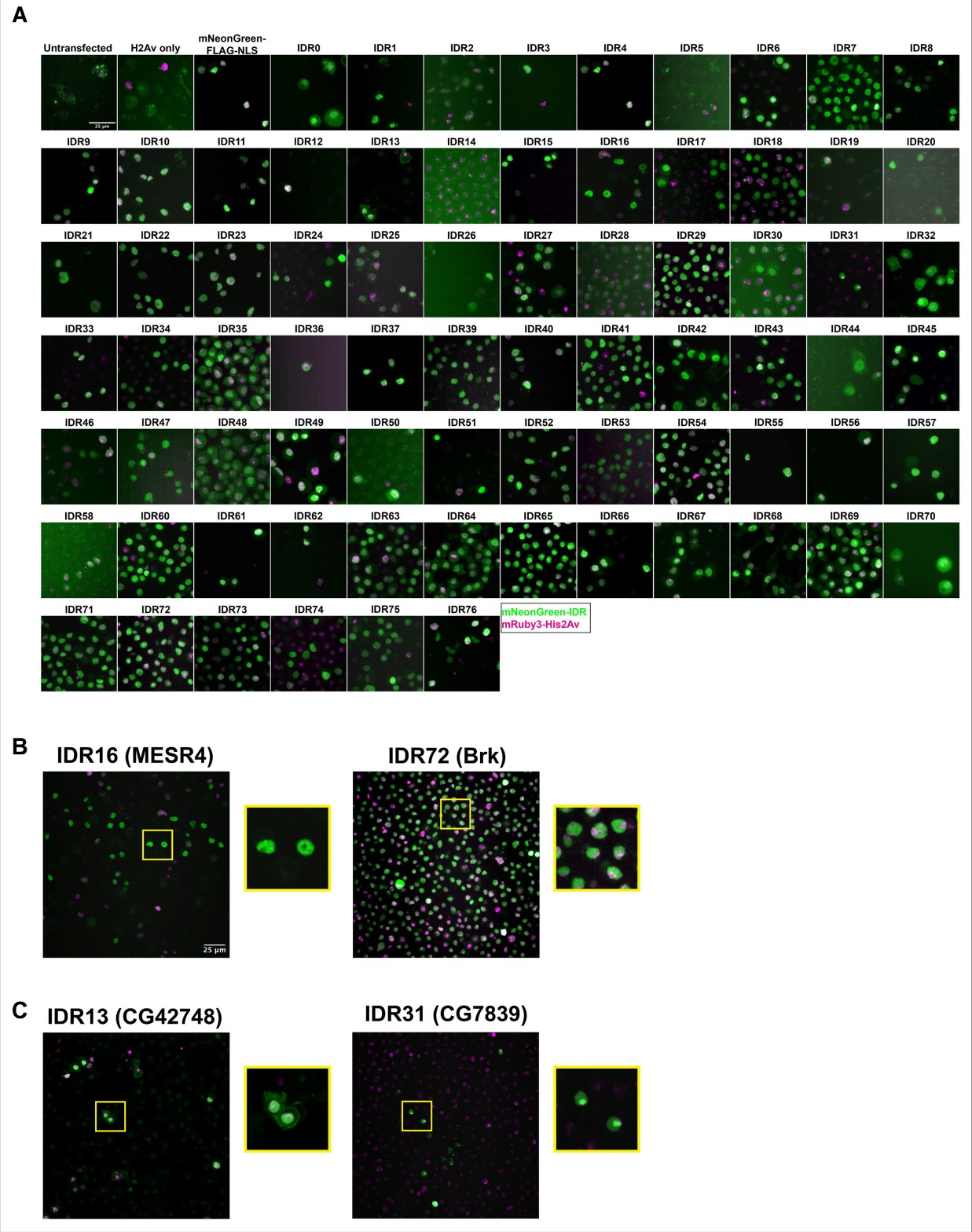

**Figure 2.** Intrinsically disordered region (IDR) imaging screen. (**A**) Representative images from each Schneider-2 (S2) cell line in the imaging screen. Untransfected controls were transfected with the p8HCO methotrexate resistance plasmid and maintained alongside experimental cell lines. His2Av only was transfected with p8CHO and pCopia-mRuby3-His2Av. All other cell lines were transfected with the pCopia-mNeonGreen-tagged IDR indicated + p8 CHO + pCopia-mRuby3-His2Av. The mNeonGreen-FLAG-NLS line is expressing the pCopia-mNeonGreen construct with no IDR inserted. Images were cropped to ~70 µm² for display. (**B**) Enlarged images from panel (**A**) for the IDRs from MESR4 and Brk, both of which show

*Figure 2 continued on next page*

Figure 2 continued

subnuclear clustering. (**C**) Enlarged images from panel (**A**) for the IDRs from CG42748, which localizes to both the nucleus and the plasma membrane, and CG7839 which localizes to the nucleolus and is present throughout the rest of the nucleus.

*1*. We also generated an S2 cell line expressing full-length Zelda, as a positive control for subnuclear clustering. Indeed, Zelda showed the typical clustering observed for this TF in the embryo (**Figure 3**).

Three of the nine full-length proteins, CG10321, Spps, and CG31510, showed notable subnuclear clustering relative to their corresponding IDR alone (**Figure 3**). The gene *CG10321* encodes an 855 amino acid DNA-binding protein containing a zinc finger-associated domain and a cluster of C2H2-type zinc fingers (**Gebali et al., 2019**; **Krystel and Ayyanathan, 2013**; **Letunic et al., 2015**). Though it is largely uncharacterized, it is expressed throughout embryonic development and the *Drosophila* life cycle (**Brown et al., 2014**) and is predicted to act as a transcriptional regulator (**Gaudet et al., 2011**). *Sp1-like factor for pairing sensitive-silencing (Spps)* encodes a zinc finger DNA-binding protein that has been shown to bind to Polycomb group response elements and to potentially play a role in recruiting Polycomb repressive complexes to these sites (**Brown and Kassis, 2010**). It is expressed ubiquitously in the embryo and continues to be expressed throughout the larval and adult life cycle (**Brown et al., 2014**). *CG31510* is expressed throughout the *Drosophila* life cycle (**Brown et al., 2014**) and encodes an 1150 amino acid protein that is almost entirely uncharacterized. Though it does not contain any annotated canonical DNA-binding domains, it contains two predicted C2H2-type zinc fingers (**Gebali et al., 2019**) in addition to its disordered sequence. The IDR from CG31510 appears to have nucleolar localization, which is maintained in the full-length version, in addition to bright clusters throughout the nucleus. The dramatic clustering of these full-length proteins relative to their IDRs alone suggests that amino acid sequences outside of the IDRs are necessary for the nuclear dynamics of these TFs.

## IDRs are not sufficient to drive subnuclear clustering in the embryo

Given the differential clustering between full-length TFs and their IDRs in S2 cells, we next asked whether these behaviors would be consistent in *Drosophila* embryos. We created fly lines expressing fluorescently tagged versions of the same subset of TFs for which we generated full-length expression constructs in S2 cells. For this set of TFs, we used CRISPR to tag the endogenous locus of each TF with eGFP. We reasoned that eGFP would be a more versatile tag than mNeonGreen for downstream applications that require an antibody, such as western blots or pull-downs. In parallel to tagging the endogenous locus of each TF, we generated transgenic lines expressing each IDR alone, tagged with mNeonGreen. We imaged each of the TFs and their corresponding IDRs in live nuclear cycle 14 (NC14) embryos, a developmental time point when the zygotic genome is transcriptionally active and each of the TFs in our panel are known to be expressed (**Tomancak et al., 2007**).

Similar to the results observed in S2 cells, the IDRs alone were largely homogenous within the nucleus (**Figure 4A**). In contrast to the IDRs alone, the full-length TFs show a range of subnuclear

**Table 1.** Panel of transcription factors (TFs) chosen for full-length expression constructs in Schneider-2 (S2) cells and *Drosophila* embryos.

| IDR # | FlyBase ID | TF name | IDR length (AAs) | Full length | IDR start | IDR end |
|---|---|---|---|---|---|---|
| 6 | FBpp0304504 | Da | 219 | 710 | 116 | 334 |
| 17 | FBpp0303090 | Rib | 249 | 661 | 413 | 661 |
| 18 | FBpp0071577 | CG10321 | 270 | 835 | 307 | 576 |
| 26 | FBpp0076735 | CG13287 | 245 | 461 | 1 | 246 |
| 44 | FBpp0081483 | Tgo | 251 | 642 | 392 | 642 |
| 64 | FBpp0083950 | Spps | 265 | 968 | 372 | 636 |
| 65 | FBpp0084158 | CG31510 | 260 | 1150 | 90 | 349 |
| 72 | FBpp0071007 | Brk | 264 | 704 | 1 | 264 |
| 76 | FBpp0074028 | Disco | 206 | 568 | 220 | 425 |

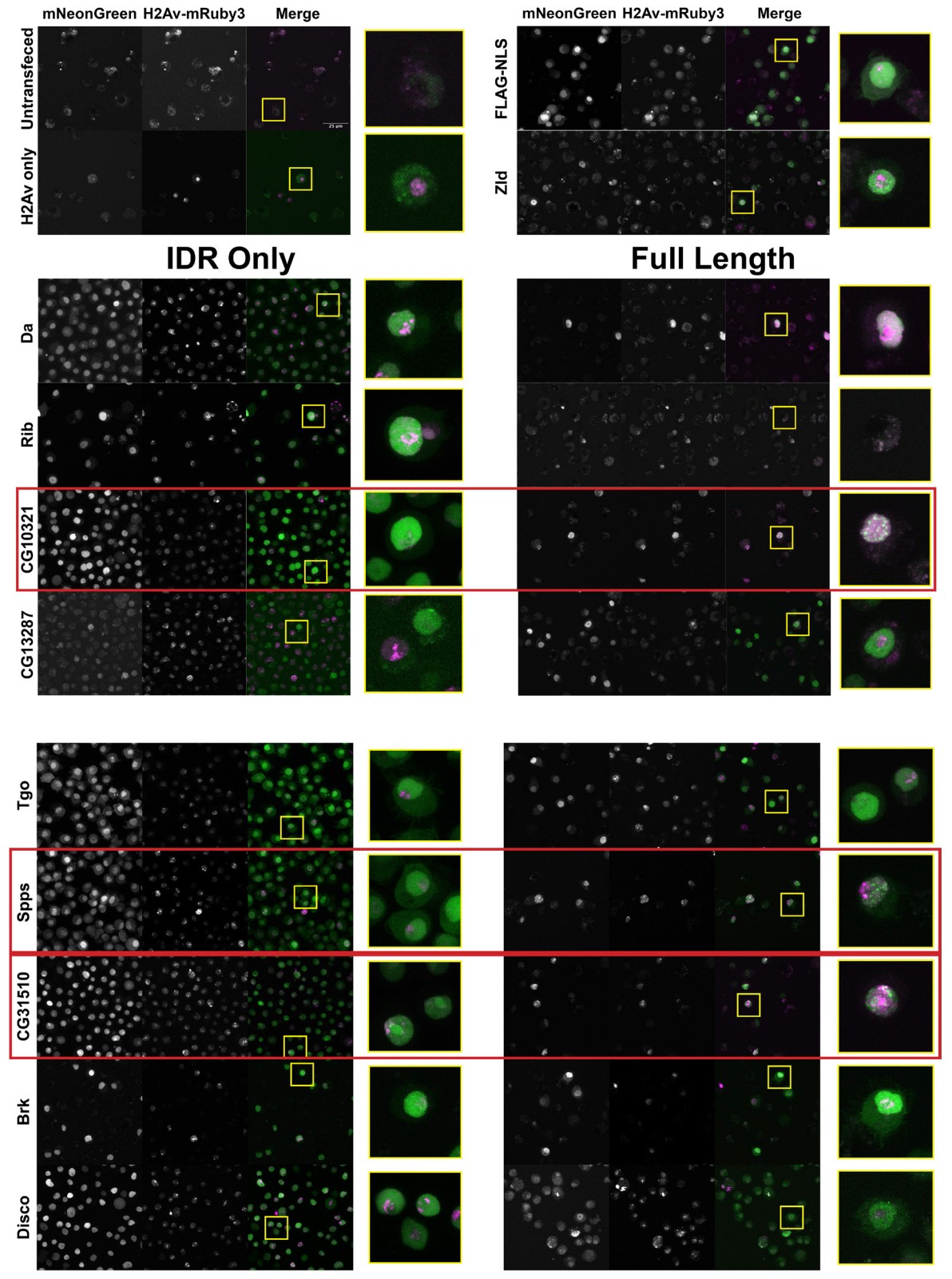

**Figure 3.** A subset a full-length transcription factors (TFs) cluster in Schneider-2 (S2) cells. S2 cell lines expressing mNeonGreen-tagged intrinsically disordered regions (IDRs) or full-length proteins and mRuby3-tagged His2Av. The top four panels indicate control cell lines. IDRs and their full-length counterparts are shown in the remaining panels. The name of the TF is indicated at the far left. IDRs alone are shown in the panels on the left and full-length proteins on the right. The TFs that show the strongest clustering are indicated with red boxes. No positively transfected cells were identified for

*Figure 3 continued*

the full-length Rib expression construct. Images are maximum intensity z-projections, and contrast was adjusted uniformly across the entire image for display.

The online version of this article includes the following figure supplement(s) for figure 3:

**Figure supplement 1.** Viterbi plots for candidate transcription factors (TFs).

localization patterns, as well as expression levels (*Figure 4B*). Three of the nine TFs – CG10321, Spps, and CG31510 – show substantial subnuclear clustering. These behaviors reproduce the findings in S2 cells, where the same three TFs clustered when the full-length protein was expressed. Da, Rib, Tgo, Brk, and Disco are largely uniform or show very fine-scale clustering. One protein, CG13287, had no visible expression, though its mRNA is expressed in a small domain on the dorsal side of NC14 embryos (*Tomancak et al., 2007*). Taken together, the imaging data from live embryos recapitulates the findings in S2 cells, namely that the IDRs identified in this study are insufficient to drive the subnuclear clustering we observe for the full-length TFs.

Interestingly, the patterns of clustering in CG10321, Spps, and CG31510 TFs are quite distinct from the clusters observed in S2 cells. In embryos, CG10321 forms bright foci that appear to localize around the edges of the nucleus. Spps, despite fairly low overall expression levels, shows many bright clusters throughout the nucleus that we might expect from a DNA-binding TF. We hypothesize that these clusters correspond to sites of Polycomb-mediated silencing. Finally, CG31510 forms bright clusters which are oriented apically in the nuclei, which are non-overlapping with the Histone 2Av signal. This is consistent with the formation of the nucleolus on the apical side of NC14 nuclei (*Falahati et al., 2016*), and therefore may recapitulate the nucleolar localization observed for both the IDR and full-length CG31510 protein in S2 cells.

In the embryo, the IDR of Brk showed one or two bright foci of subnuclear localization (*Figure 4A*). This differed from its nuclear distribution in S2 cells, where it showed small clusters throughout the nucleus (*Figure 2B*). We suspected that these foci corresponded to the HLB. The HLB forms around the histone gene cluster, where factors required for the replication-coupled transcription and processing of histone mRNAs are concentrated (reviewed in *Marzluff et al., 2008*). To test whether the Brk IDR foci corresponded to the HLB, we generated a fly line expressing an mRuby3-tagged *multi sex combs (mxc),* a homeotic gene and structural component of the HLB (*White et al., 2011*). We imaged embryos expressing both the fluorescently tagged IDR from Brk and Mxc and found very strong co-localization within the HLB (*Figure 4—figure supplement 1*). This specialized localization of the Brk IDR indicates that IDRs can play a role in concentrating factors within the nucleus. However, our screen demonstrates that this is not a common function of IDRs in general. The Brk TF is not known to be associated with the HLB, and the full-length Brk protein does not appear to have this localization pattern (*Figure 4B* and *Figure 5*). It is therefore unclear why this particular IDR is addressed to the HLB. However, as the Brk IDR is the only IDR in our panel that also contains the DNA-binding domain, the HLB localization could also be the result of DNA binding to the histone locus.

Rib, Brk, Disco, and CG13287 are expressed at low levels or in spatially restricted patterns in the embryo that make detailed imaging challenging. In order to bypass this issue, we also generated transgenic versions of all of the full-length TFs, expressed from the *nanos* promoter and tagged with mNeonGreen. The transgenic TFs are expressed ubiquitously in the embryo and with a brighter fluorophore than eGFP, allowing for more effective imaging. These transgenes enable us to capture the potential behaviors of these proteins outside of the restrictions of their endogenous context. Despite all of the constructs being expressed from the same promoter, the transgenic TFs showed a range of nuclear expression levels that are similar to the endogenous proteins (*Figure 5*), indicating post-transcriptional regulation. The transgenic TFs reproduce the subnuclear clustering patterns of the endogenous TFs, but the brighter mNeonGreen fluorophore and ubiquitous spatial expression in the embryo allows for several additional observations. CG13287, which was not visible when tagged at its endogenous locus, shows a bright and fairly uniform distribution within nuclei. Rib shows fine-scale clustering that was not visible when the protein was tagged at the endogenous locus. Tgo, though still expressed at a low level, does show some degree of clustering. And finally, Brk and Disco both appear to be associated with chromatin. The range of subnuclear behaviors observed for the different TFs indicates a diversity in the binding patterns and interactions of these TFs throughout the nucleus.

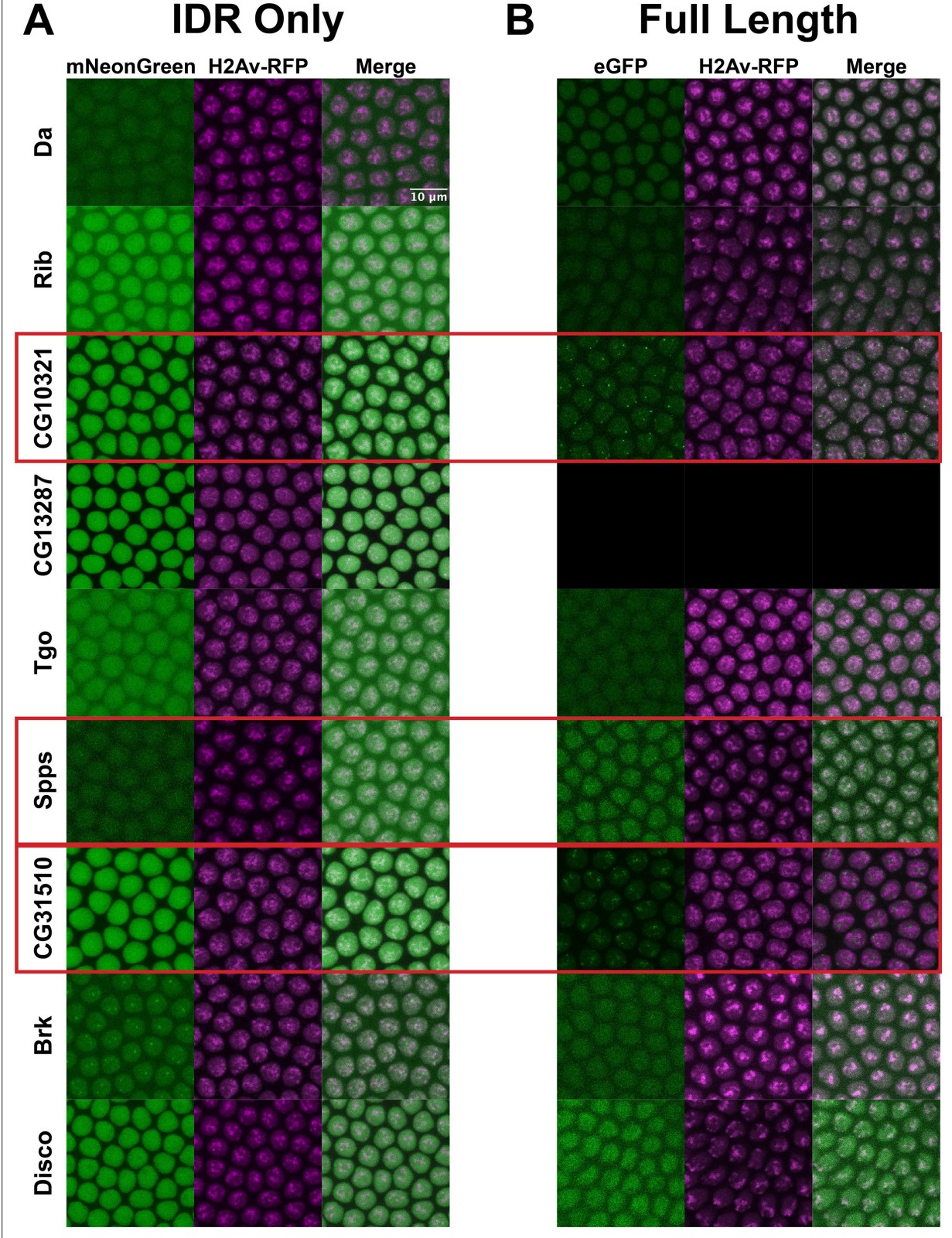

**Figure 4.** Intrinsically disordered regions (IDRs) vs full-length transcription factors (TFs) in embryos. Expression of transgenic mNeonGreen-tagged IDRs (**A**) or full-length TFs tagged at the endogenous locus with eGFP (**B**) and His2Av-RFP in nuclear cycle 14 (NC14) embryos. The name of the TF is indicated at the far left. The TFs that show the strongest clustering are indicated with red boxes. No full-length CG13287 expression was observed in embryos. Images are maximum intensity z-projections, and contrast was adjusted uniformly across the entire image for display.

*Figure 4 continued on next page*

*Figure 4 continued*

The online version of this article includes the following figure supplement(s) for figure 4:

**Figure supplement 1.** Brk localizes to the histone locus body.

## IDRs are not necessary to drive subnuclear clustering in the embryo

Though we have demonstrated that IDRs are not sufficient for TF clustering, it remains possible that they are still contributing to the localization of full-length TFs through inter- or intramolecular interactions. To test the necessity of the IDRs for the observed subnuclear localization, we generated uniformly expressed transgenic lines expressing each of the TFs with their IDRs deleted, tagged with mNeonGreen. Four of the TFs, Rib, CG10321, CG13287, and Tgo, had severely reduced nuclear expression when the IDR was deleted (*Figure 6*). This is possibly the result of the loss of a nuclear localization signal that resides within the IDR sequence. However, the most strongly clustering TFs in the set, CG10321, Spps, and CG31510, maintain their subnuclear clustering behavior with the IDRs

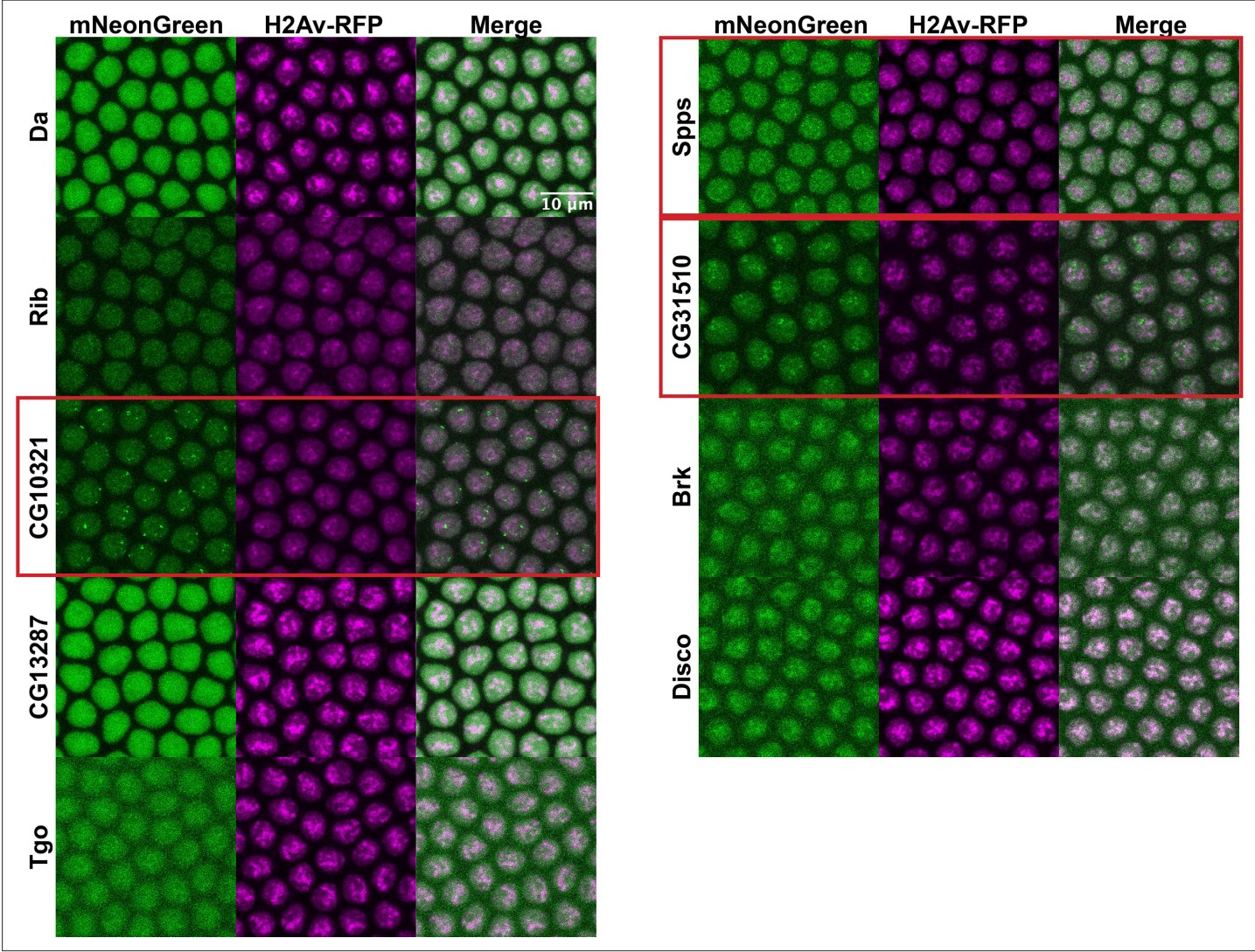

**Figure 5.** Transgenic full-length transcription factors (TFs). Nuclear cycle 14 (NC14) embryos expressing transgenic mNeonGreen-tagged full-length TFs and His2Av-RFP. The name of the TF is indicated at the far left. Images are maximum intensity z-projections, and contrast was adjusted uniformly across the entire image for display.

The online version of this article includes the following figure supplement(s) for figure 5:

**Figure supplement 1.** Subnuclear localization patterns of intrinsically disordered region (IDR) deletion constructs are uniform across the embryo.

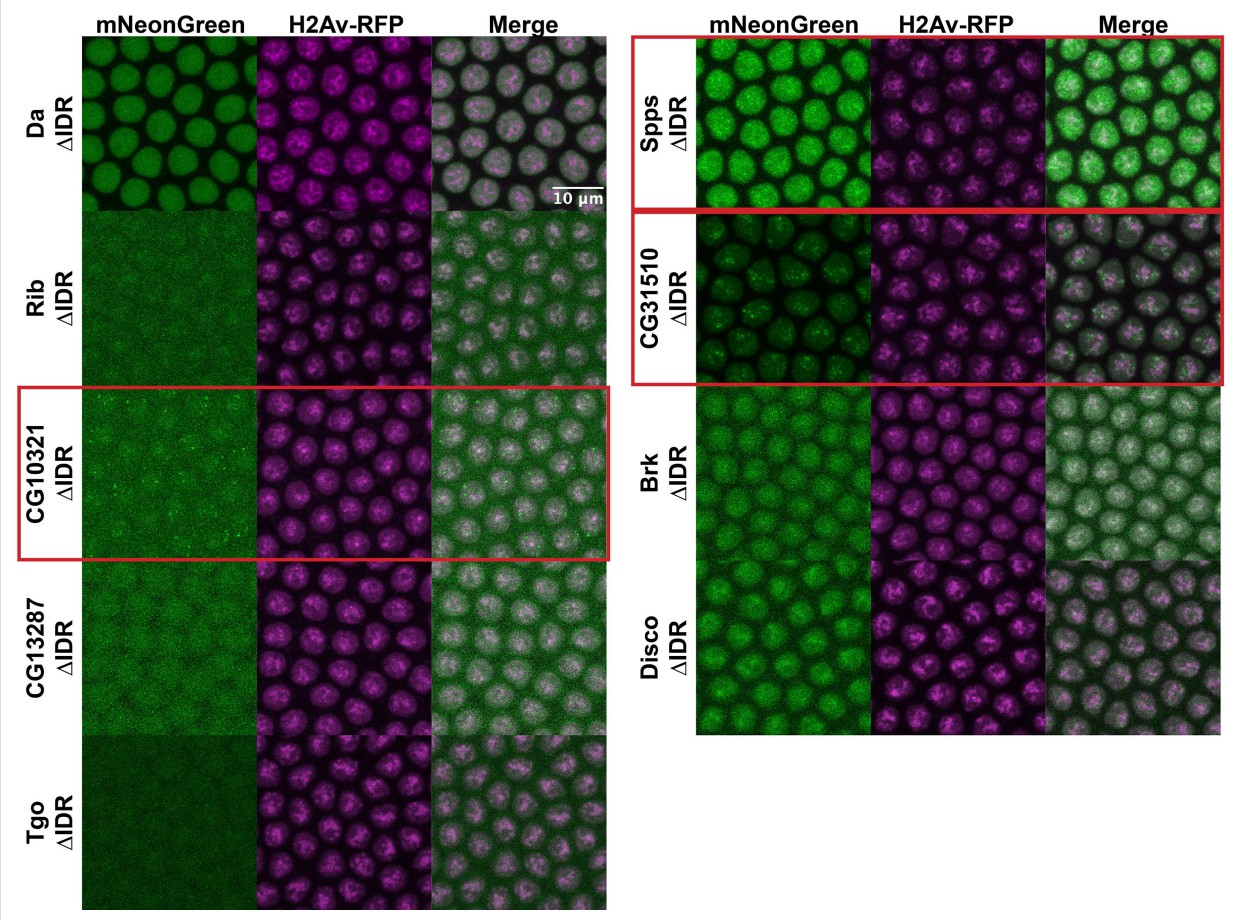

**Figure 6.** Intrinsically disordered region (IDR) deletions do not affect transcription factor (TF) localization. Nuclear cycle 14 (NC14) embryos expressing transgenic mNeonGreen-tagged TFs with IDR deletions and His2Av-RFP. The name of the TF is indicated at the far left. Images are maximum intensity z-projections, and contrast was adjusted uniformly across the entire image for display.

deleted. Indeed, even with drastically reduced expression, the clusters in CG10321 are still visible. Brk and Disco also retain their chromatin localization with their IDRs deleted. These results demonstrate that the IDRs are not necessary for the clustering behavior of the full-length TFs.

To rule out the possibility that our IDR deletion constructs were interacting with unlabeled endogenous TF in order to cluster within the nucleus, we took advantage of the patterned expression of Disco and Brk. Both are expressed in a specific subset of nuclei and absent elsewhere in the embryo, with Brk expressed in a ventrolateral stripe (*Figure 5—figure supplement 1A*) and Disco in the posterior (*Figure 5—figure supplement 1*). We imaged the Brk and Disco IDR deletion constructs in live embryos, inside and outside of the endogenous expression domains – dorsal nuclei for Brk (*Figure 5—figure supplement 1B*) and anterior nuclei for Disco (*Figure 5—figure supplement 1D*). We observe no significant differences in subnuclear distribution in nuclei where the endogenous proteins are absent, suggesting these TFs can maintain their subnuclear localization with their IDRs deleted and without interaction with the endogenous full-length protein.

## Discussion

The role of IDRs in nuclear organization and transcription has been well documented. IDRs have indeed been implicated in driving liquid-liquid phase separation, promoting the formation of a diverse array of nuclear condensates (*Elbaum-Garfinkle et al., 2015*; *Nott et al., 2015*; *Strom et al., 2017*). There is also growing evidence that IDRs are involved in the formation of transcriptional condensates at promoters (*Boija et al., 2018*; *Chong et al., 2018*; *Cho et al., 2018*; *Sabari et al., 2018*; *Wei et al., 2020*). More recently, however, the role of IDR-mediated phase separation in transcription activation

has been called into question, given the highly dynamic nature of TF hub formation and the fact that true liquid-liquid phase separation is generally observed at concentrations far above those observed for nuclear TFs (*Chong et al., 2022*; *Ferrie et al., 2022*; *McSwiggen et al., 2019*; *Trojanowski et al., 2022*). Here, we have shown that IDRs are neither necessary nor sufficient to drive the subnuclear clustering of TFs, results which demonstrate that clustering is not an emergent behavior of IDRs, and broadly rule out IDRs as major contributors to subnuclear clustering of TFs into functional hubs.

Previous work from our lab has shown that TFs form high-concentration, highly dynamic hubs that associate with active transcription in embryonic nuclei (*Mir et al., 2018*). Given the high proportion of disordered, low complexity amino acids found in TFs, we sought to delve deeper into the structure-function relationship between intrinsic protein disorder and subnuclear hub formation. A functional imaging screen of 75 IDRs from *Drosophila* TFs revealed that almost all (73/75) were insufficient to drive protein clustering on their own in cultured S2 cells. This finding was further supported by the fact that several full-length TFs do cluster in the same cells. These results were recapitulated by imaging experiments with fluorescently tagged endogenous TFs and transgenic IDRs in living embryos. Finally, we demonstrated that IDRs are dispensable for the subnuclear clustering that we observe, as TFs with their IDRs deleted still follow the same clustering patterns as their full-length counterparts.

The IDR from the Brk TF is an interesting exception, as the IDR alone does appear to cluster within nuclei. We identified a 264 amino acid IDR at the N-terminus of the Brk protein. Contained entirely within this IDR is the Brk DNA-binding domain, which has been shown to be unstructured in the absence of DNA (*Cordier et al., 2006*). While the subnuclear localization of the full-length TF is not affected by the deletion of the IDR, the IDR alone has unique localization patterns in both S2 cells and embryonic nuclei. In S2 cells, it is one of only two IDRs that has a non-uniform distribution, with visible small, bright foci throughout the nucleus. In embryonic nuclei, this fine-scale clustering is not apparent, but instead the IDR localizes to the HLB. We hypothesize that the clustering of the IDR in S2 cells is a result of the intact DNA-binding domain within the IDR binding to DNA, while the HLB localization could also be the result of non-specific DNA binding. Why the IDR does not localize to the HLB in S2 cells remains unclear. However, the coupling of the IDR and DNA-binding domain in this protein raises the possibility that TF hub formation is more dependent on DNA binding than IDR-IDR interactions. It would therefore be of interest to determine if the three most strongly clustering TFs in our study, CG10321, Spps, and CG31510, lose their ability to cluster in the absence of their DNA-binding domains.

In this work, we used bulk confocal imaging in living cells and embryos to conduct a broad survey of TF and IDR dynamics. This approach, while conducive to a high-throughput investigation of many proteins, may be limiting in the types of nuclear dynamics that we can observe. Several studies have utilized advanced imaging techniques and single molecule tracking to characterize the influence of IDRs on TF mobility and demonstrated rapid turnover of transcriptional hubs (*Chong et al., 2018*; *Cho et al., 2018*; *Cisse et al., 2013*; *Mir et al., 2018*; *Mir et al., 2017*), and identified stable and transient populations of TF molecules (*Chong et al., 2022*; *Garcia et al., 2021*). These findings leave open the possibility that the clustering that we are observing in this study are hubs that are stably bound to chromatin, and our bulk imaging approach is insufficient to observe more transient interactions between IDRs alone. More advanced imaging techniques such as single molecule tracking, with higher time resolution, could be helpful in subsequent studies to better characterize the subtleties of the subnuclear dynamics of the TFs in our data set.

The finding that IDRs are not the primary drivers of subnuclear clustering leaves us with remaining questions about their contribution to TF function. Recent studies investigating the interplay between IDRs and adjacent folded domains in the same protein have shown IDRs can adopt different conformations depending on the intramolecular interactions that they have with adjacent folded domains, and their conformation is influenced by their position within a protein (*Taneja and Holehouse, 2021*). Recent evidence has also shown that IDRs flanking folded binding domains can influence their binding affinity in protein-protein interactions, and may modulate the specificity of interaction networks (*Karlsson et al., 2022*). IDRs have also been shown to play roles in modulating DNA-binding specificity and affinity of TFs via intramolecular interactions with DNA-binding domains (*Baughman et al., 2022*; *He et al., 2019*; *Krois et al., 2018*; *Liu et al., 2009*). These studies suggest that the effects of a given IDR may be specialized and context-dependent, rather than generalizable across all IDRs. As evidence continues to emerge about the complexities of IDR functions in TFs and beyond, it is clear

that many questions remain that will require further study and characterization to elucidate the full scope of their role in transcriptional regulation.

## Materials and methods
### Identifying IDRs in the proteome

The *Drosophila melanogaster* genome (release 6.25) was downloaded from FlyBase (*Larkin et al., 2021*) as a GFF file. The genome file was parsed using Python3 to identify genes, mRNAs, and proteins, and to extract Pfam (*Gebali et al., 2019*) and SMART (*Letunic et al., 2015*) protein domain annotations. The FlyBase FASTA file for translations of protein coding transcripts from *D. melanogaster* (release 6.25) genome was parsed for amino acid sequences using the SeqIO tool from Biopython (*Cock et al., 2009*).

To predict disorder from the proteome, IUPred was run locally using the IUPred2A Python script (*Erdős and Dosztányi, 2020*; *Mészáros et al., 2018*). Three types of predictions were run: 'long' (predicts disorder spanning a minimum of 30 consecutive amino acids), 'short' (predicts smaller regions of disorder, such as residues missing from structural databases), and 'glob' (predicts disorder while reducing noise from small disordered sequences located within otherwise structured regions).

An HMM was initiated using the Python package pomegranate (*Schreiber, 2018*), taking the IUPred 'long' output as the input to the model. Starting states of the model were defined as state s0 (structured) or s1 (unstructured), and were determined from the distribution of IUPred scores in regions annotated as SMART or Pfam domains vs. the IUPred scores outside of annotated domains. Discrete continuous domains of protein disorder were defined using a Viterbi prediction, in which disordered domains were given a value of 1 and structured domains were given a value of 0. The domains identified by the model were filtered to exclude those containing Pfam/SMART domains and further filtered to include proteins classified DNA-binding TFs (*Shazman et al., 2014*). They were then filtered by size, to include only IDRs between 200 and 400 amino acids in length. This resulted in a list of 130 IDRs from the proteome. These were further filtered for IDRs encoded within a single exon, which could be amplified by PCR from the genome, giving a final set of 78 IDRs. The final list is shown in *Supplementary file 1*.

### Generating stable S2 cell lines

From the IDRs identified in the *Drosophila* proteome, we filtered by size for ease of expression and cloning, and identified 130 IDRs between 200 and 400 amino acids. Of this subset, we identified 78 IDRs that were encoded by a single exon. Using this list, we designed primers to amplify each of the 78 IDRs from OregonR genomic DNA (*Supplementary file 1*). Using ligation-independent cloning, we cloned each IDR sequence into a pCopia expression vector (*Parker et al., 2019*) containing an N-terminal mNeonGreen or mRuby3 fluorescent tag, followed by a FLAG tag, and with an SV70 NLS sequence downstream of the IDR. We were unable to amplify three of the IDR sequences (#38, 59, and 77), so the final library contained 75 fluorescently tagged IDRs.

To generate stable integrations of the fluorescent IDR constructs, we co-transfected with p8HCO, which confers methotrexate resistance (*Rebay et al., 1991*). *Drosophila* S2 cells were maintained as an adherent culture at 27°C in ESF 921 Media supplemented with 1% FBS, 100 units/mL penicillin, 200 µg/mL streptomycin, and 0.25 µg/mL amphotericin B (Gibco Antibiotic-Antimycotic). 96-well plates were seeded with 100 µL of S2 cells at a density of 1.5×10e6 cells/mL. After 24 hr, the media was replaced and the cells were co-transfected with three plasmids: 1 of the 76 pCopia-mNeonGreen-IDR plasmids, pCopia-mRuby3-Histone2Av, and p8CHO (100 ng of each plasmid per transfection), using Effectene transfection reagent (QIAGEN Catalog #301425). The cells were incubated with the transfection mixture at 27°C for 48 hr, and the media was then replaced and supplemented with methotrexate at a concentration of 0.1 µg/mL. The cells were maintained at 27°C and the methotrexate media was replaced 2× per week for 5 weeks, after which time the transfected cells contained significant populations of mRuby3 and mNeonGreen positive cells.

Stable cell lines expressing mNeonGreen-tagged full-length TFs were generated using the same cloning and transfection procedures described for the IDR library. Primer sequences are available in *Supplementary file 1*.

## High-throughput imaging of S2 library

The IDR S2 cell lines were prepared for imaging by resuspending and transferring to glass bottom 96-well microplates (Corning). Plates were imaged on an Opera Phenix high-throughput confocal microscope (Perkin Elmer). Cells were imaged using a 63×1.15 NA water objective, and z-stacks were collected with 0.5 µM slice intervals.

## Imaging full-length TF cell lines

The full-length TF and corresponding IDR S2 cell lines were expanded to six-well plates and prepared for imaging by resuspending and transferring to 27 mm glass bottom cell culture dishes (ThermoFisher). Cells were imaged on a Zeiss (Germany) LSM 800 scanning confocal microscope using 488 nm and 561 nm lasers in a 101.4 µM$^2$ window. Cells were imaged using a Plan-Apochromat 63×1.40 NA oil-immersion objective, and z-stacks were collected with 1 µM slice intervals.

## Generation of transgenic fly lines

For the IDR expression constructs, DNA fragments encoding IDRs or full TFs were amplified from genomic DNA extracted from a single OregonR fly. Primers for amplification are available in *Supplementary file 2*. The IDR fragments or full-length TF sequences were then cloned via Gibson Assembly into the pMRS-213 vector (Michael Stadler, unpublished), containing a *nanos* promoter, N-terminal mNeonGreen tag, and *alpha-tubulin* 3'UTR. For ease of amplification, the longer coding sequences to express the full-length TFs from transgenic constructs were amplified from the pCopia-mNeonGreen expression constructs used for S2 cell expression. The full-length sequences were then cloned by Gibson Assembly into pMRS-213.

The IDR deletion sequences were synthesized by GenScript (Piscataway, NJ, USA) and cloned into pMRS-213. Both the IDR and IDR deletion constructs Rainbow Transgenic Flies, Inc (Camarillo, CA, USA) and injected into stock *y[1] v[1] P{y[+t7.7]=nos-phiC31\int.NLS}X; P{y[+t7.7]=CaryP}attP40* (Bloomington Stock #25709) for site directed integration into *attp40* site.

Full-length TF reporter lines were generated using CRISPR/Cas9 mutagenesis with homology directed repair (HDR). sgRNA targeting sequences were annealed and cloned into the pMRS-1 vector (*Mir et al., 2018*). HDR constructs were designed containing an N-terminal FLAG tag, eGFP, and a flexible linker, flanked by ~1 kb homology arms. The homology sequence directed the insertion of the tag in frame with the start codon of each gene. The HDR sequences were cloned into pUC19 via Gibson assembly. For two genes, *tgo* and *CG13287*, tagging at the N-terminus was unsuccessful. The HDR constructs were re-designed to insert a flexible linker, FLAG tag, and eGFP between the N-terminus of the encoded protein and the stop codon. The HDR plasmids were pooled with gene-specific sgRNA guide plasmids and pCFD3-ebony as a visible co-CRISPR marker (*Kane et al., 2017*), and sent to Rainbow Transgenic Flies for injection into embryos expressing Cas9 in the germline. After injected larvae hatched into adults, individual flies were crossed to balancer lines for the appropriate chromosome, each also carrying a *TM3* balancer marked with *ebony*. The F1 progeny were screened for the presence of the *ebony* mutant phenotype, and *ebony* mutants were crossed again to an appropriate balancer stock to generate stable fly lines. After the F2 crosses were established, the *ebony* F1 parents were sacrificed for PCR genotyping with primers to amplify the junction between eGFP and the flanking HDR homology sequence. Positive eGFP-tagged lines were further characterized by amplification of the target region using primers outside of the homology arms, followed by Sanger sequencing to confirm the correct insertion of the fluorescent tag. Guide RNA sequences, primers for generating homology arms, and primers for screening CRISPR insertions are available in *Supplementary file 3*.

## Confocal imaging in living embryos

Embryos were collected from apple juice plates and sorted by stage in halocarbon 27 oil. Appropriately staged embryos were mounted in halocarbon 27 oil between a coverslip and gas permeable membrane. Confocal images were collected on a Zeiss LSM 800, using 488 nm and 561 nm lasers in a 101.4 µM$^2$ window. Embryos were imaged using a Plan-Apochromat 63×1.40 NA oil-immersion objective, and z-stacks were collected with 0.5 µM slice intervals.

## Acknowledgements

We thank Max Staller for comments on the manuscript, Matthew Parker and Christi Abbate for guidance and assistance with cell culture, Michael Stadler for sharing reagents, discussing experiments, and commenting on the manuscript, Holli Weld for help with cloning IDR library, and members of the Eisen lab for helpful feedback and discussion. We thank Deepa Sridharan of the High-Throughput Screening Facility (HTSF) at UC Berkeley. This work was performed in part in the QB3 HTSF, which provided the Perkin-Elmer Opera Phenix microscope. CEH was funded by an American Cancer Society Postdoctoral Fellowship (133547-PF-19-004-01-CCG). MBE is an investigator with the Howard Hughes Medical Institute.

## Additional information

### Funding

| Funder | Grant reference number | Author |
|---|---|---|
| American Cancer Society | 133547-PF-19-004-01-CCG | Colleen E Hannon |
| Howard Hughes Medical Institute | | Colleen E Hannon<br>Michael B Eisen |

The funders had no role in study design, data collection and interpretation, or the decision to submit the work for publication.

### Author contributions

Colleen E Hannon, Conceptualization, Resources, Data curation, Formal analysis, Funding acquisition, Validation, Investigation, Visualization, Methodology, Writing – original draft, Writing – review and editing; Michael B Eisen, Conceptualization, Software, Formal analysis, Supervision, Funding acquisition, Project administration

### Author ORCIDs

Colleen E Hannon ⓘ https://orcid.org/0000-0002-4402-8107
Michael B Eisen ⓘ http://orcid.org/0000-0002-7528-738X

Reviewer #1 (Public Review): https://doi.org/10.7554/eLife.88221.2.sa1
Reviewer #2 (Public Review): https://doi.org/10.7554/eLife.88221.2.sa2
Author Response: https://doi.org/10.7554/eLife.88221.2.sa3

## Additional files

### Supplementary files

• Source code 1. Python code used to generate the HMM IDR predictions, provided as a Jupyter Notebook.

• Supplementary file 1. It contains identifying information, amino acid, and DNA sequences of all intrinsically disordered regions (IDRs) expressed in Schneider-2 (S2) cells in *Figure 2*. This file also contains all primer sequences used to generate the IDR library, and primer sequences used to generate the full-length protein constructs shown in *Figures 2 and 3*.

• Supplementary file 2. It contains sequences of primers used to generate transgenic *Drosophila* lines expressing intrinsically disordered regions (IDRs) or full-length transcription factors (TFs) in *Figures 4–6*.

• Supplementary file 3. It contains sequences of primers used to generate CRISPR tagging constructs, sgRNA sequences, screening primers to identify CRISPR insertions, and a list of injected fly lines to produce the endogenously tagged transcription factors (TFs) shown in *Figure 4* and *Figure 4—figure supplement 1*.

• MDAR checklist

## Data availability

The python code used to generate the HMM IDR predictions referenced in *Figure 1* is supplied as a Jupyter Notebook in *Source code 1*.

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
