## [Editor Report · eLife assessment]

The manuscript addresses a **fundamental** question: are IDRs responsible for subnuclear clustering of transcription factors? A screen of 75 IDRs yielded **convincing** evidence that IDRs are rarely sufficient for subnuclear clustering, while the experimental design and data analysis provided limited evidence for the authors' claims regarding transcription factor clustering.

---

## [Referee Report · Reviewer #1 (Public Review)]

Recent works have documented the observation that transcription factors (TFs) and other transcriptional machinery proteins, e.g., Pol2 and mediator, can form high-concentration clusters at target genes within the nucleus and such behavior plays an important role in transcriptional activation. It is also well-established that the intrinsically disordered regions (IDRs) within many of the transcriptional regulators can undergo multivalent protein-protein interactions, which can lead to phase separation under certain conditions. It is thus thought that the IDRs are essential drivers of the clustering behaviors of transcriptional regulators. However, direct proof of this hypothesis remains missing. To fill this gap, Hannon and Eisen conducted a survey of the subnuclear localization of 75 IDRs derived from *Drosophila* TFs. They found that many full-length TFs but not IDRs alone form subnuclear clusters. They also did not detect a change in the clustering of TFs after deleting IDRs. Based on these data, they concluded that IDRs are unlikely to be the primary molecular drivers of the clustering phenomenon observed during transcription.

This study tackles an interesting question related to transcriptional regulation and IDR behaviors. The subnuclear distribution of many *Drosophila* TFs and TF IDRs measured in this work provides a valuable resource for future studies of these TFs and IDRs. The authors' finding that the distribution of an IDR alone is distinct from a full-length TF containing the IDR is new and informative though not surprising. Protein-chromatin binding is known to be stoichiometric with a clear structural basis, which is likely stronger than IDR-IDR interactions that are known to be weaker and more transient. Thus, adding a chromatin-binding capability to an IDR of interest indeed likely significantly affects the distribution of the IDR. Despite being a natural hypothesis based on existing knowledge, it was not previously verified until this current work that systematically compared TF and TF IDR distributions. However, the authors' other conclusion that deleting the IDR from a TF does not affect the TF's clustering behavior is not fully supported. This is because the change of TF's clustering behavior due to IDR deletion, if any, is likely quantitative (decreasing) instead of qualitative (completely disappearing), due to the fact that protein-chromatin binding is stronger than IDR-IDR interactions. This work lacks quantitative characterization of TF clusters before and after IDR deletion. The spinning disk confocal microscopy used here likely does not provide the necessary spatial resolution to quantitatively characterize TF clusters, which often have dimensions near or below the diffraction limit of light.

---

## [Referee Report · Reviewer #2 (Public Review)]

This work by Hannon and Eisen focuses on the sequence and structural features of transcription factors (TFs) that dictate their sub-nuclear localization. The authors test the hypothesis that intrinsically disordered regions (IDRs) in TFs are drivers of subnuclear localization and clustering by first identifying IDRs in the *Drosophila* proteome using a novel approach and then expressing a subset of IDRs from TFs important during the development of an early embryo. The authors then perform an extensive and high-throughput imaging screen in S2 cells and *Drosophila* embryos and find that subnuclear clustering does not occur when IDRs are expressed alone but happens frequently in full-length TFs, even sometimes without the IDRs. A significant strength of the study is the extensive amount of imaging data that support well the conclusions in the paper. A potential weakness is that the conclusions are based on qualitative analysis only; the work would be strengthened considerably if the authors could provide quantification that allows the reader to distinguish clearly between a homogenous distribution and clustering of TFs. The work tackles an important functional question regarding IDRs in TFs and is of high relevance to the field. There is an impressive amount of data that generally support the conclusion of the paper, which is that IDRs are insufficient to drive TF clustering in the nucleus. The manuscript is very well written, pleasing to read, and easy to follow. This work advances the field considerably, providing valuable mechanistic insights into transcription.

---

## [Author Response]

We would like to thank the reviewers for their comments on the manuscript. The primary concern that they raised is that the imaging data are largely qualitative. This is a fair assessment, and we agree that a careful quantitative characterization of TF clustering with and without IDRs using high resolution imaging would provide valuable insight that would extend our findings. Our goal for this study was to conduct a high level survey of IDR localization, for which we believe a qualitative overview was sufficient. We hope that this work can serve as a useful foundation for future studies of the complex roles that IDRs play in TF function.